# Effects of Rho-Associated Kinase (Rock) Inhibitors (Alternative to Y-27632) on Primary Human Corneal Endothelial Cells

**DOI:** 10.3390/cells12091307

**Published:** 2023-05-03

**Authors:** Gary S. L. Peh, Francisco Bandeira, Dawn Neo, Khadijah Adnan, Yossa Hartono, Hon Shing Ong, Sacha Naso, Anandalakshmi Venkatraman, José A. P. Gomes, Viridiana Kocaba, Jodhbir S. Mehta

**Affiliations:** 1Tissue Engineering and Cell Therapy Group, Singapore Eye Research Institute, Singapore 169856, Singapore; 2Singhealth Duke-NUS Ophthalmology & Visual Sciences Academic Clinical Programme , Duke-NUS Graduate Medical School, Singapore 169857, Singapore; 3Department of Ophthalmology and Visual Sciences, Federal University of São Paulo, São Paulo 04023-062, Brazil; 4Corneal and External Diseases Department, São Gonçalo Eye Hospital, Rio de Janeiro 24421-005, Brazil; 5Bioinformatics Institute, A*STAR, 30 Biopolis Street, Matrix #07-01, Singapore 138671, Singapore; 6Singapore National Eye Centre, Singapore 168751, Singapore; 7Netherlands Institute for Innovative Ocular Surgery, 3071AA Rotterdam, The Netherlands

**Keywords:** ophthalmology, cornea, corneal endothelium, primary human corneal endothelial cells, cell therapy, cell injection, regenerative medicine, corneal transplantation, Rho-associated coiled-coil protein kinase

## Abstract

(1) Rho-associated coiled-coil protein kinase (ROCK) signaling cascade impacts a wide array of cellular events. For cellular therapeutics, scalable expansion of primary human corneal endothelial cells (CECs) is crucial, and the inhibition of ROCK signaling using a well characterized ROCK inhibitor (ROCKi) Y-27632 had been shown to enhance overall endothelial cell yield. (2) In this study, we compared several classes of ROCK inhibitors to both ROCK-I and ROCK-II, using in silico binding simulation. We then evaluated nine ROCK inhibitors for their effects on primary CECs, before narrowing it down to the two most efficacious compounds—AR-13324 (Netarsudil) and its active metabolite, AR-13503—and assessed their impact on cellular proliferation in vitro. Finally, we evaluated the use of AR-13324 on the regenerative capacity of donor cornea with an ex vivo corneal wound closure model. Donor-matched control groups supplemented with Y-27632 were used for comparative analyses. (3) Our in silico simulation revealed that most of the compounds had stronger binding strength than Y-27632. Most of the nine ROCK inhibitors assessed worked within the concentrations of between 100 nM to 30 µM, with comparable adherence to that of Y-27632. Of note, both AR-13324 and AR-13503 showed better cellular adherence when compared to Y-27632. Similarly, the proliferation rates of CECs exposed to AR-13324 were comparable to those of Y-27632. Interestingly, CECs expanded in a medium supplemented with AR-13503 were significantly more proliferative in (i) untreated vs. AR-13503 (1 μM; * *p* < 0.05); (ii) untreated vs. AR-13503 (10 μM; *** *p* < 0.001); (iii) Y-27632 vs. AR-13503 (10 μM; ** *p* < 0.005); (iv) AR-13324 (1 μM) vs. AR-13503 (10 μM; ** *p* < 0.005); and (v) AR-13324 (0.1 μM) vs. AR-13503 (10 μM; * *p* < 0.05). Lastly, an ex vivo corneal wound healing study showed a comparable wound healing rate for the final healed area in corneas exposed to Y-27632 or AR-13324. (4) In conclusion, we were able to demonstrate that various classes of ROCKi compounds other than Y-27632 were able to exert positive effects on primary CECs, and systematic donor-match controlled comparisons revealed that the FDA-approved ROCK inhibitor, AR-13324, is a potential candidate for cellular therapeutics or as an adjunct drug in regenerative treatment for corneal endothelial diseases in humans.

## 1. Introduction

The corneal endothelium (CE) plays a critical role in corneal transparency through a balance between the leaky cellular barrier and the active ionic pump mechanism that maintains corneal deturgescence [1,2]. Human corneal endothelial cells (CECs) have a low capacity to regenerate within the eye [3] due, in part, to the high concentrations of TGF-ß2 and cAMP within the aqueous humor that is believed to up-regulate p27^KIP1^, preventing the CECs from proliferating [4,5]. Corneal endothelial dysfunctions, due either to a progressive or an acute loss of cells, have a negative impact on the CEC dynamic functional integrity, resulting in corneal edema, stromal scarring, and compromised visual acuity, which will eventually lead to corneal blindness [6]. 

The current treatment to restore vision involves the replacement of the diseased CE through a corneal transplant of a healthy donor corneal graft [7]. However, there is an unattainable demand for corneal transplants due to a global shortage of suitable donor graft tissue, with only 1 in 70 patients receiving corneal transplant worldwide [8]. This has stimulated research on the use of alternative treatment modalities [9]. The leading causes of corneal endothelial dysfunctions are Fuchs endothelial corneal dystrophy (FECD), and conditions that incur massive endothelial cell loss such as bullous keratopathy and corneal re-grafting secondary to endothelial failure. In early FECD disease, one option is an autologous regeneration approach, in which the diseased corneal endothelium is selectively removed to trigger the migration of healthy CECs from the paracentral cornea [10,11,12]. The surgery can be performed by the removal of the DM (Descemet stripping only (DSO)), or by associating DSO to an acellular DM transplant over the stripped area, which facilitates cellular migration (Descemet membrane transplantation (DMT)) [10,12]. It has also been shown that adjunctive treatment with topical Rho-kinase inhibitor (ROCKi) in an eyedrop formulation has the capacity to hasten cellular migration [13,14]. 

In conditions leading to massive endothelial damage or advanced FECD with widespread endothelial dysfunction, a cell-based therapy approach is needed, where the isolation of healthy CECs from donor corneas, with or without cellular expansion, is required prior to delivery into the anterior chamber [15,16]. Delivery of the cells can be conducted by a cell injection approach, or as a cell sheet on a scaffold-based carrier. Regulatory compliant trials are underway to establish the efficacy and safety of expanded CECs in the replacement of the dysfunctional endothelium, and early clinical results have shown promising improvement in visual acuity and a reduction in corneal edema [17]. 

The Rho-associated coiled-coil protein kinase is part of the serine/threonine family and acts as an effector of the Rho signaling pathway. With 65% of overall identity, two isoforms are present in the human body: ROCK-1 and ROCK-2 [18]. They are distinguished by their different amino-acid composition [19] and dynamics of dimerization [18]. These ROCK isoforms are expressed heterogeneously throughout the human body [19]. Within the eye, both kinases have been detected in the cornea (epithelial, limbal, and endothelial cells) [20], the trabecular meshwork [21], and the retina [22], with ROCK-2 found to be more prevalent [23]. The effects of ROCK activation are associated with a multitude of cellular events [23,24], ranging from actin cytoskeleton organization, cellular adhesion, cellular proliferation, cytokinesis modulation [25], and the induction of apoptosis [26]. These effects play important roles in modulating the adhesion and migration of CECs and are both significant components for corneal endothelial migration and proliferation [20]. The most common form of ROCK inhibition is achieved through the use of small molecules known to competitively bind the H/NH moiety and aromatic groups to the ROCK ATP kinase domain. Several ROCK inhibitors have been developed to inhibit the ROCK pathway signaling cascade, and these ROCK inhibitors comprise a diverse molecular scaffold with inhibitory capacity ranging from the micromolar down to the subnanomolar order of magnitude [27]. Some of these subclasses of ROCK inhibitors include pyridines, isoquinolines, benzodioxane amides, and indazoles, to name a few [28,29]. Along with specific cell-type responses to ROCK, they provide versatility in the form of potential treatment options for different medical pathologies. For example, ROCK inhibitors have been reported to modulate cell growth in the oncogenesis of breast [30] and pancreas [31] tissues. Additionally, The use of ROCKi promoted the cellular regeneration in degenerative diseases such as supranuclear palsy [32]. Clinically, there are reports of the ophthalmic use of ROCK inhibitors in the form of eyedrops. One such eyedrop, Ripasudil, has been approved to treat open-angle glaucoma and ocular hypertension in Japan [33,34]. More recently, Rhopressa, a ROCKi compound consisting of Netarsudil, was approved by the US Food and Drug Administration to reduce intraocular pressure through increased trabecular outflow [35,36]. 

In the cornea, reports have shown that the use of ROCK inhibitors enhanced endothelial wound closure [9], as well as ex vivo cellular migration [37,38]. They have also been used to promote the propagation of isolated primary CECs for in vitro expansion [20,39,40]. One such compound, a widely described pyridine known as Y-27632 [41,42], has been shown to promote cellular adherence, increasing cell growth, and inhibiting apoptosis of both primate CECs [43] and human CECs [39,44]. In addition, the inclusion of Y-27632 was able to increase the yield of cultured primary human CECs 2.6-fold, through both the enhancement of cellular attachment and the promotion of cell proliferation [39]. Similar results have been found with other ROCK inhibitors [45], such as Y-39983 [44]. The promising in vitro reports on primary cultures of CECs and their safe clinical use for glaucoma [46] have since extended application of ROCK inhibitors to the treatment of the corneal endothelium as an off-label indication. Indeed, the topical application of Ripasudil has been shown to enhance autologous regeneration of the corneal endothelium [47]. 

In this study, we investigated various classes of ROCK inhibitors for their effects on primary CECs cultured using an established dual media culture system [48]. With regulatory compliance in mind, we specifically included the ROCK inhibitors that have been reported for clinical ophthalmic use in our assessment. The following ROCK inhibitors were selected: Ripasudil (K-155), Netarsudil (AR-13324), and its active metabolite (AR-13503) [46]. Cellular effects of these ROCK inhibitors were assessed using donor-matched cultures of primary CECs with the Y-27632 set as baseline control. Subsequently, the ROCKi with the most favorable effect on the CECs was brought forward for additional comparative studies, alongside Y-27632, on its effects on the cell proliferation of CECs, as well as the ex vivo regeneration of the corneal endothelium. 

## 2. Materials and Methods

### 2.1. Research-Grade Human Corneoscleral Tissue

A total of 32 pairs of research-grade human cadaver donors were procured for this study through either Lions Eye Institute for Transplant and Research (Tampa, FL, USA), or Saving Sight (Kansas City, MO, USA), each with written consent from the next of kin, and adhering to the principles outlined in the Declaration of Helsinki. For in vitro optimization studies, procured donor corneas were from younger donors ranging from 4 to 35 years old (serial number 1 to 22); whereas for ex vivo studies, older donors (>50 years old) were procured. All cornea pairs had endothelial cell density (ECD) of at least 2000 cells/mm^2^ that were deemed unsuitable for transplantation (Table 1). Corneoscleral tissues were preserved in Optisol-GS (Bausch and Lomb, Rochester, NY, USA) at 4 °C until they were processed, generally within 14 days of preservation. 

### 2.2. ROCK Inhibitor Compounds

To investigate and compare the effects of different ROCK inhibitors on human CECs, various ROCKi compounds were obtained. The compound Y-27632 was purchased from Miltenyi Biotec GmbH (Cologne, Germany). Research grade standard of the FDA-approved ophthalmic compound known as Netasurdil (AR-13324), along with its active compound, AR-13503, and Verosudil (AR-12286) [49] were obtained from Aerie Pharmaceuticals Inc. (Bedminster, NC, USA). The powder form of the clinically approved ROCKi eye drop, Ripasudil (K-115 hydrochloride dihydrate), and Y-39983-HCL were purchased from Afirmus Biosource (Selleckchem, Houston, TX, USA). Other ROCK inhibitor compounds of various classes assessed within this study included an amniofurazan, an amide, an indazole, and a benzodiazepine (all GSK, London, England, UK). All ROCKi compounds used in this study have been summarized in Table 2.

### 2.3. In Silico Experiments

#### 2.3.1. PDB Screening for ROCK Proteins

Structures of ROCK-1 and ROCK-2 were obtained from the Protein Data Bank (PDB) according to the highest resolution found for either protein, using a cluster cut-off of 0.51. Redundant structures were excluded based on clustering the co-crystallized ligands by their structural similarity. 

#### 2.3.2. ROCKi Docking

In silico docking of several ROCK inhibitor molecules from different classes (Isoquinolines, Aminofurazan, Benzodiazepine, Indazole and Amide) and Y27632 was performed with the Glide application [50] within the software package Schrödinger Release 2018-2 (Schrödinger, LLC, New York, NY, USA, 2018). The receptors were prepared with Protein Preparation Wizard and ligands with tautomeric and protonation states were generated with LigPrep. For docking, extra precision (XP) mode was used, with van der Waals scaling of ligand atoms by 0.8. Finally, the ligand scores were clustered into the respective classes of each ROCK inhibitor’s individual docking, and their means were compared to Y27632 (control) and amongst themselves.

### 2.4. In Vitro Experiments

#### 2.4.1. Primary Culture of Human Corneal Endothelial Cells 

Primary human CECs were isolated using a two-step “peel and digest” approach [51] and propagated using the dual media approach as described previously [52]. Briefly, isolated CECs were first established in a cornea endothelial maintenance/stabilization medium (M5-Endo; Human Endothelial-SFM supplemented with 5% EquaFetal) overnight. Subsequently, CECs were cultured in a proliferation medium (M4-F99; Ham’s F2/M199, 5% EquaFetal, 20 μg/mL ascorbic acid, 1X ITS, and 10 ng/mL bFGF) to promote their proliferation. Once cell growth reached approximately 80–90% confluent, M5-Endo medium was re-introduced to the culture for at least two days before being sub-cultured using TrypLE Select (TS) dissociation. Dissociated CECs were plated at a seeding density of at least 1 × 10^4^ cells per cm^2^ onto surfaces pre-coated with FNC coating mixture for further expansion. All cell cultures were propagated within a humidified atmosphere at 37 °C with 5% CO_2_. 

#### 2.4.2. Cellular Viability Assay Using xCelligence 

All experiments carried out using the xCelligence real-time cell analyzer for the comparative studies of the ROCK inhibitors to determine their effects, were performed with a minimum of at least three biological repeats (n = 3). These experiments involved the use of E-Plates (ACEA Biosciences, San Diego, CA, USA) with gold-microelectrodes fused to their culture surface. As the CECs adhered, presence of cells at the electrode–solution interface impedes electron flow. The impedance is measured by the integrated software and is converted into arbitrary cell indices for comparative analysis. Here, the negative controls in each experiment were its respective donor-matched primary CECs in M5-Endo media without any ROCKi supplementation, whereas the positive control group used in the analysis was M5-Endo media containing 10 µM Y-27632 [39]. Primary human CECs dissociated into single cells at the second passage were seeded at a density of 3.0 × 10^4^ cells/cm^2^ in each well of the E-Plate that had been pre-coated with FNC coating mix (AthenaES, Baltimore, MD, USA). The CECs were left to be stabilized over 24 h in M5-Endo without any ROCKi. On the following day, the CECs were exposed independently to M5-Endo medium containing the respective ROCKi (see Table 2) for 24 h. This phase was denominated the “drug” phase. Subsequently, the ROCKi-supplemented medium was withdrawn from all cultures, replaced with fresh M5-Endo medium, and allowed to incubate for additional 24 h in a “recovery” phase. It should be noted that this step is critical in determining if any detrimental effect observed during the “drug” phase was sustained, which would suggest possible cellular toxicity. In turn, if the negative effect was reversible, some form of recovery in the CECs, as gauged by cellular impedance, will be observed. These steps have been depicted as a schematic (see Figure 1A). Electrical impedance readings of the adhered CECs, measured as arbitrary cell index (CI), were recorded throughout the experiment and used as a quantitative measure to compare the effect of each compound on the CECs. For analysis, CI readings were taken at two timepoints, specifically at T = +24 (24 h after exposure to ROCKi; Figure 1A) and at T = +48 (24 h following recovery phase; Figure 1A).

#### 2.4.3. Click-iT Cell Proliferation Assessment

The proliferation rates of primary CECs were assessed using the EdU incorporation Click-iT cell proliferation assay (ThermoFisher Scientific, Waltham, MA, USA) as per the manufacturer’s instructions. Two ROCK inhibitors, AR-13324 and AR-13503, were assessed for their capacity to enhance proliferation of CECs, with two concentrations (100 nM or 1 µM for AR-13324 and 1 µM or 10 µM for AR-13503). Donor-matched CECs with no ROCKi added served as negative control, whereas CECs with Y-27632 added served as positive control. Briefly, cultured CECs, passaged using TS, were seeded onto FNC-coated glass slides at a density of 5 × 10^3^ cells per cm^2^ and maintained in M5-Endo for 24 h (Day 1). On the second day (Day 2), the medium was switched to each respective condition, and cells were cultured for another 24 h. On the third day, cells were incubated in M4-F99 containing 10 mM of EdU for 24 h. Subsequently, samples were rinsed once with PBS before they were fixed in freshly prepared 4% PFA for 15 min at room temperature. Next, Samples were rinsed twice with 3% BSA in PBS and were incubated in 0.5% Triton X-100 in PBS for 20 min at room temperature for blocking and permeabilization. Incorporated EdU was detected by fluorescent-azide-coupling Click-iT reaction where samples were incubated for 30 min in the dark with a reaction mixture containing Click-iT EdU reaction buffer, CuSO_4_, azide-conjugated Alexa Fluor 488 dye, and reaction buffer additive. Following that, samples were rinsed with 3% BSA before incubating in 5 µg/mL Hoechst 33,342 for 10 min at room temperature in the dark. Finally, samples were washed twice in PBS and mounted in Vectashield containing 4,6-diamidino-2-phenylindole (DAPI), (Santa Cruz Biotechnology, Dallas, TX, USA). Labelled proliferative cells were examined under a Zeiss Axioplan 2 fluorescence microscope (Carl Zeiss, Oberkochen, Baden-Württemberg, Germany). At least 250 nuclei were analyzed for each experimental condition.

### 2.5. Ex Vivo Wound Model

To assess corneal endothelial wound closure after ROCKi exposure, we chose a previously described approach with two types of wounds created in ex vivo cultured corneas: (1) DM-stripped to expose bare posterior stroma (peeled wound) and (2) CEC-denuded DM (scratched wound) [37]. Briefly, the outlines of two circles were lightly marked on the corneal endothelial surface with a 3 mm disposable biopsy punch (World Precision instruments, Sarasota, FL, USA). A gap of approximately 2 mm was left in between these two marked zones. Next, scraped wounds were created by gently removing endothelial cells with a custom-made silicone soft-tip probe (ASICO, IL; Item: AS-7661). Finally, peeled wounds were created by a continuous curvilinear descemetorhexis (CCD). The CCD involved the initial creation of a DM tear at the center of the circle, followed by extending the tear in a continuous and curvilinear manner until a complete circle of DM had been peeled off.

Following wound creation, donor corneas were maintained in an ex vivo culture setup in M5-Endo medium supplemented with Y-27632 (10 µM; OD) or AR-13324 (1 µM; OS) throughout the course of the three-week study as previously described [37]. The culture medium supplemented with the respective ROCKi was replenished every other day. All corneas were maintained within a controlled environment at 37 °C with carbon dioxide concentration set at 5%.

#### 2.5.1. Image and Processing 

Specimens were imaged immediately following initial wound creation and every week until Day 21. In order to enhance the visualization of the created wounds, each cornea was briefly incubated for 30 s in a 0.2% Trypan Blue solution (TBS; Sigma-Aldrich Corp., Singapore) as described previously [37]. Following immersion in TBS, areas with intact, non-damage corneal endothelium remained translucid, whereas both the peeled and scraped areas exhibit blue dye due to TBS uptake. It should be noted that peeled area was more readily distinguished from the scraped area by the presence of marked descemetorhexis margins [37]. Images were obtained using the Nikon SMZ1500 stereomicroscope and the Nikon DSFi 1-L2 high-definition color camera (Nikon Instruments), with backlit illumination provided by a halogen light source located beneath the microscope stage. All images were checked, and the camera software performed automatic white balancing prior to the acquisition of each image. Image analysis was performed with the color threshold tool using the ImageJ software (National Institutes of Health, Bethesda, MD, USA) as follows (Appendix A). Briefly, after converting the images to BandW 8-bit format, a baseline wound area was measured using the image taken immediately after wound creation. The subsequent measurements of the wounded areas were obtained by adjusting the color threshold in such a way as to encompass where CECs were absent (i.e., appeared dark). Maximum endothelial recovery referred to the area of the wound that achieved negative staining (i.e., appeared white) by the end of the experiment, measured as a percentage of the total wound area obtained with the initial image. The endothelial recovery rate was calculated as maximum endothelial recovery divided by 21 (total length of the experiment) and measured as percentage recovery per day.

#### 2.5.2. Alizarin Red Staining 

Donor-matched corneas (n = 3) were randomly selected for Trypan Blue/Alizarin red staining following the end of ex vivo culture. Alizarin red solution (0.5%) was freshly prepared on the day of staining. The Alizarin red powder (Merck, Darmstadt, Germany) was dissolved in distilled water. The pH was titrated to 4.5 before filtering using a syringe pump filter. Each of these corneas was first stained for 3 min in buffered 0.2% TBS, and the cornea was then immediately stained in freshly prepared Alizarin red solution. The specimen was then washed for 60 s in a wash buffer prior to wet mounting and examination with a Zeiss Axioplan 2 microscope for wide-field images and a Zeiss Confocal Microscope (Zeiss, Obkerkochen, Germany), for high-magnification images.

### 2.6. Statistical Analysis

Prism 6.0 (GraphPad, Inc., San Diego, CA, USA) was used for data analysis; comparisons in cellular proliferation were carried out using Tukey’s Test for multiple comparisons. Kruskal–Wallis with Dunn post hoc corrections for multiple comparisons were applied to detect intergroup differences in endothelial migration. The Friedman test was used to analyze the endothelial wound recovery rate within each wound. All numerical data obtained were expressed as mean ± standard deviation (SD) unless otherwise stated. The results were deemed statistically significant when *p* < 0.05 was achieved. 

## 3. Results

### 3.1. In Silico Protein Binding Assay

A total of 23 ROCK structures were found in the PDB. The maximum and minimum resolutions were 3.4 Å and 2.93 Å, respectively. Seven ROCK-I and two ROCK-II non-redundant structures were selected for the binding assay. Out of 46 compounds tested (20 isoquinolines, 15 aminofurazan, 6 benzodiazepine, 4 indazoles, and 1 amide), 34 presented a significantly higher docking score for ROCK-1, when compared to Y-27632 (*p* < 0.0001). All ROCKi classes presented a stronger mean docking score than Y-27632 (*p* < 0.0001). The frequency of compounds presenting highest docking score was higher in the isoquinoline, aminofurazan, and benzodiazepine classes for ROCK-I; and in isoquinolines and amides for ROCK-II (Appendix AA). The top ten compounds that presented the highest mean docking scores for ROCK-I and II are shown in Appendix AB. The isoquinoline class represented 70% of the drugs within the top ten highest docking scores, with three compounds presenting a docking score stronger than −12. There were no significant differences among ROCK inhibitors other than Y-27632.

Interestingly, in silico molecular docking simulation showed that the majority of the molecules evaluated, specifically from the isoquinoline, benzodiazepine, and amide classes, had higher binding strength for ROCK-1 and ROCK-2 than Y-27632 (Appendix AB). In silico molecular docking simulation was performed, coupling isoforms found for AR-13324 and Y-27632 inhibitors in the PDB to high-resolution ROCK proteins. All of the AR-13324 molecules tested had a higher docking score for ROCK-1 and -2 than Y-27632. In addition, PDB molecules from the isoquinoline, benzodiazepine, and amide classes also showed superior mean docking scores than Y-27632 isoforms (Appendix AB).

### 3.2. Effects on Cellular Impedance from Exposure to Various ROCK inhibitors Vary in Human CECs

The impedance readings for each set of ROCKi-treated CECs were first normalized internally, where the recorded CI were normalized at each time-point to the CI of the donor-matched CECs that were treated with a control ROCKi Y-27632 at 10 µM, and where this normalized CI was set to 1.0 (Figure 1B–J, red dotted line). For each ROCKi, at least three different concentrations were used in this initial assessment, and it was evident that their effects differed greatly. Exposure of AR-13324 to CECs at 100 nM resulted in a slight increase in cellular impedance compared to Y-27632 control, but concentration of 10 µM and 30 µM were observed to be detrimental to the CECs, an effect that appeared to be irreversible (Figure 1B). Indeed, we observed that primary CECs became non-viable following the exposure of 10 µM of AR-13324 (Appendix A). Interestingly, the concentration of 1 µM did not show any detrimental effect on the CECs (Appendix A). For AR-13503, exposure of 100 nM to CECs resulted in greater cellular impedance, although the variances between donors were relatively high, as seen by the high standard of deviation (Figure 1C). Interestingly, both 10 µM and 30 µM appeared to negatively impact the CECs (Figure 1C; 24 h). However, these effects were not toxic to the CECs (Appendix A), and the observation was reversed following ROCKi withdrawal, as seen by the recovering of their respective CIs (Figure 1C; 48 h). The CI of CECs exposed to AR-12286 at both 100 nM and 10 µM appeared to be comparable to Y-27632. Conversely, at 30 µM, it appeared to weaken slightly before recovering after the removal of the ROCKi (Figure 1D). For Y-39983, cellular impedance at 100 mM was comparable to Y-27632. At both exposures of 10 µM and 30 µM, each of their cellular impedance was negatively affected. Although close to full recovery was observed in CECs exposed to 10 µM of Y-39983, impedance remained poor in CECs that were exposed to 30 µM of Y-39983 (Figure 1E). Next, the CECs treated with Ripasudil (K-115) were only comparable to Y-27632 at the concentration of both 10 µM and 30 µM. Finally for the four different classes of ROCKi assessed, all were comparable to Y-27632 at 100 nM and 10 µM (Figure 1G–J), as well as for both indazole (G-1) and Amide (G-3) at 30 µM. Interestingly, exposure to 30 µM of Aminofurazan (G-2) appeared to exert a non-recoverable detrimental effect on the CECs (Figure 1H), whereas impedance of cells treated with 30 µM of Benzodiazepine showed full recovery (Figure 1J). 

Hence, the cellular impedance, as gauged by xCelligence, revealed a noticeable positive effect even when CECs were exposed to AR-13324 and AR-13503 at much lower concentrations than Y-27632. Therefore, both of these ROCK inhibitors were further assessed on their proliferative effect on primary CECs, and subsequently, AR-13324 was directly compared to Y-27632 with an ex vivo model of endothelial wound recovery. 

### 3.3. AR-13324 and AR-13503 Increased Proliferation Rates of Isolated Human CECs

The intracellular Edu incorporation showed increased proliferation rates across all CECs isolated from the five donors while cultured in M4-F99 supplemented with either of the two ROCK inhibitors (Figure 2). Specifically, the addition of AR-13324 at both the concentrations of 1 µM and 0.1 µM showed a similar increase in proliferation rates in a donor-to-donor comparison, with cells grown in 10 µM of Y-27632. In contrast, compared with Y-27632-supplemented CECs, the addition of AR-13503 showed a greater enhancement of proliferation rates within each donor-matched comparison across all five donors (Figure 2).

### 3.4. Corneal Endothelial Wound Recovery (Ex-Vivo)

Using pairs of donor-match corneas, an ex vivo corneal endothelial wound recovery model enabled the comparison between recovery of the CE after treatment with Y-27632 vs. AR-13324. Vital staining with trypan blue demonstrated a significant reduction in the mean wound area for peeled and scraped wounds compared to the baseline in both groups (Figure 3A,B). The respective final wound areas were of 51.35% ± 39.57% (*p* < 0.01) and 16.31% ± 9.92% (*p* < 0.001) for the AR-13324 group; and 62.14% ± 40.75% (*p* < 0.05) and 18.42% ± 21.04% (*p* < 0.001) for the Y-27632 group. The weekly mean wound healing rate (WHR) was significantly higher for scraped compared to peeled wounds in both groups at Day 7 (58.2% ± 22.8% vs. 24.9% ± 26.4%, AR-13324, *p* < 0.05; 51.4% ± 26.9% vs. 8.1% ± 8.7%, Y27631, *p* < 0.05), and it progressively decreased for both groups as the corneal endothelial sheet became more confluent (Figure 3C–F). There was no significant difference between the AR-13324 and Y-27632 groups regarding WHR or the final healed area.

### 3.5. Effects of Donor Age on Endothelial Migration

There was an apparent association between increasing age and decreased maximum and daily endothelial migration rates for peeled and scratched wounds. This trend was similar with both ROCK inhibitors and far more prominent for peeled compared to scraped wounds. Interestingly, the donor age had a stronger correlation with a delay in the wound healing for scraped wounds in corneas treated with Y-27632 than ones supplemented with AR-13324, although this observation was not statistically significant (Figure 4A, R^2^ = 0.87, *p* = 0.33 vs. R^2^ = 0.06, *p* < 0.99; respectively). Conversely, the donor age strongly correlated with delayed wound healing for peeled wounds of both Y-27632 and AR-13324 (Figure 4B, R^2^ = 0.98, vs. R^2^ = 0.93, *p* < 0.05).

### 3.6. Alizarin Red Staining of Ex Vivo Corneas

Alizarin red staining showed that exposure to AR-13324 (Figure 5A) is comparable to that of Y-27632 (Figure 5B). Specifically, recovery observed of the peeled areas (Figure 5A^†^,B^††^) covered a smaller area than the recovery of cells in the scrapped areas (Figure 5A^§^,B^§§^), and were mostly restricted to the adjacent to the wound edge. The endothelial mosaic near the acellular zones were heterogeneous, with cells that were relatively larger and less hexagonal when compared to cells in the non-wounded areas (Figure 5A^‡^,B^‡‡^). 

## 4. Discussion

Rho-kinase proteins play a role in many cellular events in different cell types [23]. With regard to human CECs, they have been shown to improve cell yield in primary culture protocols [39,40,53] and hasten endothelial wound closure in animal models [9,20,54,55]. In this study, we have demonstrated that various classes of ROCK inhibitors evaluated had comparable or better performance for both cellular attachment and proliferation of primary human CECs compared to the ROCKi, Y-27632. This is not surprising, as shown through the in silico protein docking assay which showed that Y-27632 had a weaker docking score than most of the compounds evaluated (Appendix A). In our in vitro studies, primary human CECs cultured with the different ROCKi classes presented a 10–30% increase in impedance readings when compared with cells incubated with Y-27632. This is expected, as ROCK inhibitors are known to increase focal adhesion by phosphorylating myosin light chains, myosin light chain phosphatase, and LIM kinase, regulating the formation of actin stress fibers’ assembly and cell contraction, modulating the light myosin chain [56]. Moreover, donor-matched CECs cultured in all of the ROCK inhibitors evaluated were able to recover after incubation at increasing concentrations up to 10 µM, retaining both cellular viability and morphological features. It is well established that cell adhesion is of utmost importance in achieving successful primary cell cultures [57], and our results are in line with previous reports of approximately 30% improved adhesion [20,39] and non-toxic effects of Y-27632 when limiting concentration to under 30 µM [39,43,53,58]. 

The incorporation of ROCK inhibitors in the cultivation of primary human CECs or as promoter of corneal endothelial wound healing is relatively new. It should be noted that the majority of the studies that described the effectiveness in the usage of ROCK inhibitors for primary human CECs have been conducted using the compound Y-27632 [15,20,39,45,59], which is neither FDA nor GMP approved for clinical use. For clinical ophthalmic use, Ripasudi is an FDA approved ROCKi for the treatment of glaucoma [60]. It has been reported to possess similar efficacy comparable to Y-27632 [9]. Others have shown, through in vivo experimental outcomes and case reports involving endothelial wound recovery, that the optimal dose for Ripasudil should be at least twice the dosage recommended for glaucoma treatment [13,61,62]. In addition, Ripasudil has been reported to induce rosette-like changes to the corneal endothelial cells when applied in normal corneas [63], and was associated with multiple side effects including blepharitis, bowel disorders, and hyperemia, leading to its discontinuation [64]. 

Conversely, Netarsudil (AR-13324) has been FDA approved, and with the availability of GMP culture formulation for the growth of primary human CECs [52], it could potentially be incorporated into culture protocols for clinical purposes and as an enhancer of autologous regenerative treatments. Thus, it was only fitting that we would select this drug for the subsequent series of experiments to compare its effect on cellular attachment and proliferation of primary CECs with that of Y-27632. To this end, studies to assess the adherence and mitotic effects of AR-13324 in comparison to Y-27632 were carried out using expanded donor-matched primary human CECs to negate known donor-to-donor variation. We observed that the primary cells attached slightly better over Y-27632 as gauged by their cellular impedance, but when applied at a lower concentration of 100 nM. Interestingly, AR-13324 appeared to be cytotoxic to the cultured cells when exposed at a concentration of 10 µM and above, as the CI did not recover following the withdrawal of AR-13324 (Figure 1A; Figure 3). Next, improved cellular proliferation over untreated controls, as assessed using Click-IT EDU labeling, was evident when donor-matched primary CECs were grown in an M4-F99 medium supplemented with either Y-27632 or AR-13324. Whether 1 µM (*p* = 0.93; Figure 2) or 0.1 µM (*p* = 0.17; Figure 2) of AR-13324 used in the study were comparable to the proliferation rate of using 10 µM of Y-27632, it has been suggested that the mechanistic action of ROCK inhibitors drives cyclin D, p27, and Cdk2 activation, which, in part, promoted the proliferation of the cultured primary CECs [46]. However, the proliferative effects of ROCK inhibitors in cultured CECs have been a subject of controversial debate [43,65]. Whilst some groups have shown evidence of the use of ROCK inhibitors in promoting cell proliferation [39,44,58], others indicated the absence of such effect [20,65]. We have previously characterized the proliferative effect of Y-27632 on cultured primary CECs and have shown clear evidence that different donor-derived primary CECs behaved differently with and without the presence of ROCK inhibitors [39]. Similarly, in this study, all five donor-matched CECs generally responded with enhanced proliferation rate when treated with ROCK inhibitors, although it should be noted that disparity between the rates of proliferative increase between different donors was observed. 

In this study, in addition to comparing Y-27632 and Netarsudil in a series of donor-matched studies, we also evaluated the active metabolite of Netarsudil known as AR-13503 [46], and assessed its capacity to enhance cellular attachment and cellular proliferation of CECs. Similar to AR-13324, AR-13503 increased cellular attachment of CECs when supplemented at 100 nM. Interestingly, supplementation of AR-13503 at 10 µM and 30 µM resulted in a drop of their cellular index. However, the observed effect is not as toxic to the primary CECs, as the CI of the cells recovered following the recovery phrase. For cellular proliferation, donor-match cells also responded positively to AR-13503 exposure (Figure 2; Appendix A). We observed exceptionally high proliferation rates in two of the donor-matched samples in media supplemented with both 10 µM and 1 µM of AR-13503. At the concentration of 10 µM of AR-13503, comparative proliferation rates of the donor-matched cells were statistically significant when comparisons between untreated CECs (*p* < 0.001); 10 µM Y-27632 (*p* < 0.01); 1 µM AR-13324 (*p* < 0.01); and 0.1 µM AR-13324 (*p* < 0.05) were made. At the concentration of 1 µM of AR-13503, statistical significance was achieved when the proliferation rates were compared against untreated CECs (*p* < 0.05). A recent study reported a lower dose of AR-13503 to be effective at greater than 1000 times than Y-27632, and suggested the efficacy of this lower dosage to be highly specific for the effects against ROCK [66]; it is currently assessed as an implantable in subjects with neovascular age-related macular degeneration or diabetic macular edema (NCT03835884). Indeed, the results observed in the current study suggests that when used at a higher concentration of 10 µM, AR-13503 may be a suitable molecule for the purpose of corneal endothelial cell expansion protocols and cellular therapeutics, and it is currently the subject of further investigation. 

For the endothelial wound recovery study, we have selected only corneas above 60 years old, which encompass the majority of the patients eligible for DM removal and adjunct with ROCKi treatment for FECD [67]. This decision was also based on a previous study that demonstrated corneas from younger donors, where the use of Y-27632 prevented premature culture failure and extended the viability of older corneas (>60 yo) from 3.67 ± 1.15 days to 6.0 ± 1.41 days [37]. In this study, we were able to maintain the cornea ex vivo for over 20 days with both Y-27632 as well as AR-13324, and we were able to observe remarkable wound recovery in all of our samples (Figure 3). We did not find any statistical differences in the total healed area and WHR between corneas treated with the two ROCK inhibitors, AR-13324 or Y-27632, for scraped or peeled wounds. For both ROCKi groups, the mean total WHR for peeled wounds was approximately 40% slower than the scraped ones, an observation in line with previous reports [37,68,69] in which the lack of the DM ensued a 13–20% slower regeneration in similar ex vivo wound models [37,68], along with a significantly delayed corneal edema resolution in animal studies [68,69]. 

The donor age also seemed to impact the endothelial healing dynamics for both types of wound. Previously, Soh et al. demonstrated that Y-27632 slowed the decline of wound healing significantly in corneas older than 50 years [37]. Similar to our findings (Figure 4), they have also shown that endothelial migration in peeled wounds of older corneas were found to be worse than in the scraped samples. Additionally, ROCKi supplementation greatly accentuated the differences between the endothelial migration rates in favour of the scraped wounds. Our results corroborated with the observation of Soh et al. [37] in that surgeons should carefully consider the age of patients under evaluation for regenerative treatment for corneal endothelial dysfunction with adjunct ROCKi, as well as the technique for removal of the diseased endothelium (with or without DM replacement). 

It should be noted that the ex vivo experiment is limited by a confounding factor that must not be ignored. Unlike the endothelial wound healing experiments conducted on animals, TGF-ß is absent from both the ex vivo culture system and the in vitro experimental studies. TGF-ß is a constituent of the aqueous humor, which is a known inhibitor of the proliferation of primary CECs [5]. Moreover, TGF-ß has been implicated as a key player in EMT signaling and wound healing for several corneal cell types, including keratocytes and CECs [70,71,72]. The absence of TGF-ß could favor cell regeneration and overestimate the clinical potential of ROCKi treatment in our settings. However, there was no significant difference between treatments with Y-27632 and AR-13324, making further studies in animals questionable. In addition, there is some anecdotal evidence on the clinical efficacy of Descemet stripping only (DSO) for FECD, reported with and without the addition of ROCK inhibitors [14,73,74], where approximately 35% of patients that were submitted to DSO without ROCKi-adjunctive treatment persisted with opaque corneas [12]. Concurrently, a randomized clinical trial with 18 patients showed that only a single patient that underwent DSO along with Ripasudil treatment failed to recover endothelial function [14]. Interestingly, in one interventional case series with 13 patients undergoing DSO, 3 patients that failed to recover endothelial cell function were able to be salvaged by introducing Ripasudil drops [61]. While the exact criteria for prescribing topical instillation of ROCKi eye drops for the preventive deterioration of CECs in chronic Fuchs cases remains unclear [75], there is growing evidence supporting the use of ROCK inhibitors as an adjuvant therapy in DSO [14,73,74,76]. 

The initial outcomes with ROCK inhibitors in the field of tissue engineering and regenerative medicine are exciting and promising. While tissue engineering of primary CEC has recently reached a breakthrough, DSO is yet controversial. Nevertheless, additional randomized controlled clinical trials similar to those of Kinoshita et al. [17] and Macsai et al. [14], with a larger sample size and longer follow-ups, should be carried out to confirm the safety and long-term efficacy of ROCK inhibitor assisted DSO. The fact that we now have a commercially available, FDA-approved, GMP-compliant ROCK inhibitor might hasten these studies and even pave the way for other applications of ROCK inhibitors. Examples of these alternative applications of ROCK inhibitors are their use as prophylactic protection against CECs damage during intra-ocular surgeries (mainly Phacoemulsification and Keratoplasty) and their improvement of the viability of the corneal endothelium during donor cornea preservation in eye banks.

## 5. Conclusions

In conclusion, we were able to demonstrate that additional ROCK inhibitor classes and compounds other than Ripasudil and Y-27632 hold promise in the treatment of corneal endothelial dysfunction with tissue engineering. Furthermore, the comparison between Y-27632 with the FDA-approved GMP-compliant ROCK inhibitor, AR-13324 (Netarsudil), showed similar outcomes with an in vitro cell culture system and an ex vivo wound regeneration model. Hence, this drug could be a candidate for cellular therapeutics or as an adjunct drug in regenerative treatments for Fuchs endothelial corneal dystrophy in humans.

## Figures and Tables

**Figure 1 cells-12-01307-f001:**
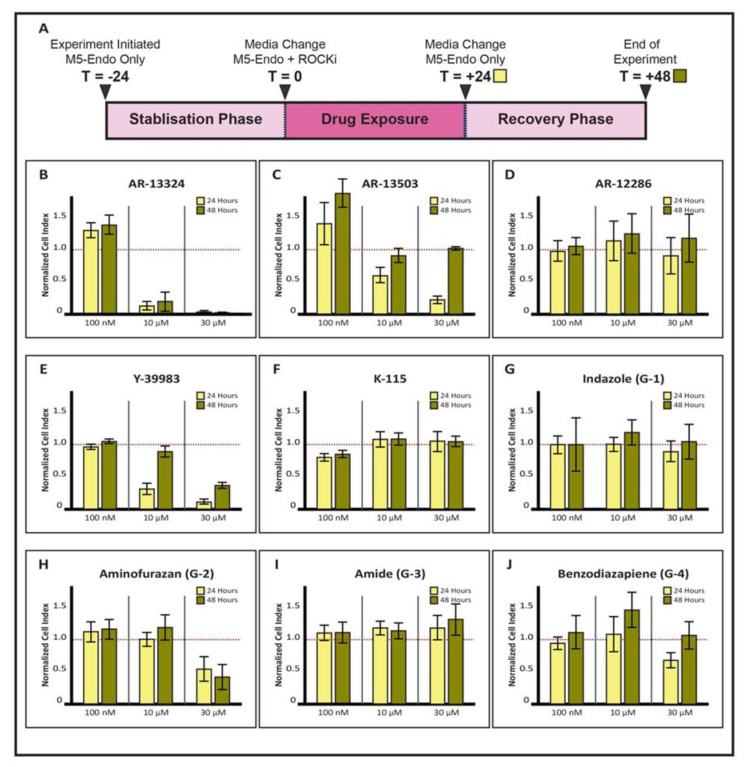
ROCK inhibitors and the attachment of primary CECs: (**A**) Schematic of the CEC viability experiment with different ROCK inhibitors; CEC Drug/Recovery attachment study with supplementation of the following ROCK inhibitors (**B**) AR-13325; (**C**) AR-13503; (**D**) AR-12286; (**E**) Y-39983; (**F**) K-115; (**G**) G-1; (**H**) G-2; (**I**) G-3; and (**J**) G-4 at the three concentrations. Bar chart showing mean and SD cell index for each ROCK inhibitor at increasing doses. All values were normalized to the cell index of Y-27632 as depicted by the red dotted line.

**Figure 2 cells-12-01307-f002:**
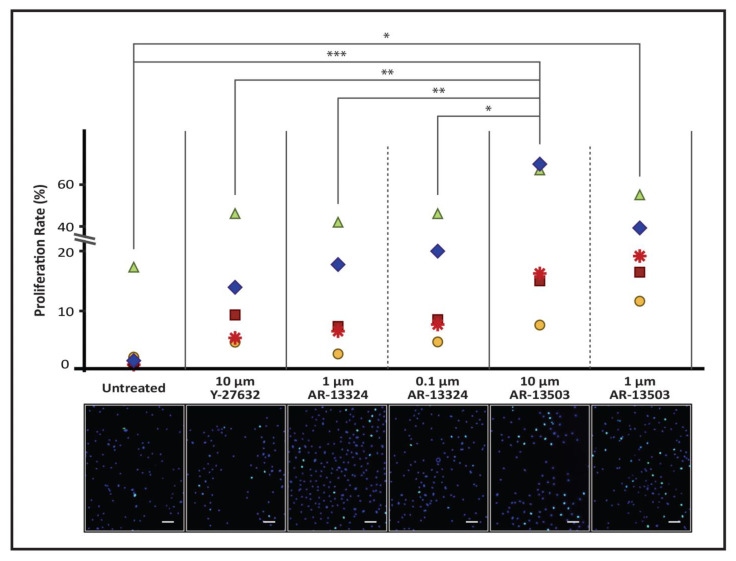
Proliferation of primary CECs treated with ROCK inhibitors. The dot plot represents the proliferation rate (%) of expanded primary CECs that were donor-matched, comparing CECs that were untreated (negative control) and those with supplementation of Y-27632 (10 µM; positive control); AR-13324 (1 μM and 0.1 μM); and AR-13503 (10 μM and 1 μM). The corresponding images below each dot plot are representative images of the Click-IT Edu proliferation assay. The merged images showed nuclei of the primary CECs counter-stained blue (DAPI), while proliferating cells were labelled with fluorescent green (Click-IT Edu-GFP). Tukey’s Test for multiple comparisons found that the proliferation rates were significantly different between (i) untreated and AR-13503 (1 μM; * *p* < 0.05); (ii) untreated and AR-13503 (10 μM; *** *p* < 0.001); (iii) Y-27632 and AR-13503 (10 μM; ** *p* < 0.005); (iv) AR-13324 (1 μM) and AR-13503 (10 μM; ** *p* < 0.005); and (v) AR-13324 (0.1 μM) and AR-13503 (10 μM; * *p* < 0.05).

**Figure 3 cells-12-01307-f003:**
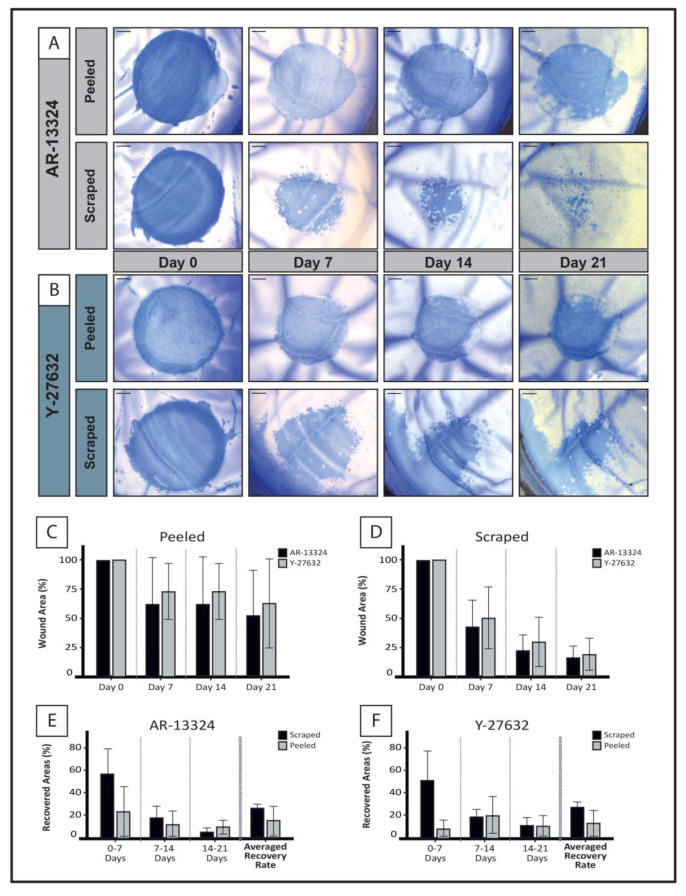
Corneal endothelial cell regeneration ex vivo wound models. (**A**, **B**) Representative images for the endothelial recovery after peeling and scraping wounds for matched pair donor corneas treated with (**A**) AR-13324 and (**B**) Y-27632. Scale bars: 500 μm. (**C**) Graph showing weekly remaining corneal wound area for peeled wounds, and at Day 21, the remaining endothelial wound area for AR-13324 was 48.65% ± 39.57% (black; *n* = 3) and, for Y-27632, 37.9% ± 40.75% (grey; *n* = 3), showing a greater but non-significant recovery for AR-13324 (*p* > 0.05). (**D**) Graph showing weekly remaining corneal wound area for scraped wounds, and similarly, Day 21 showed the remaining endothelial wound area for AR-13324 at 16.31% ± 9.92% (black; *n* = 3), and Y-27632 at 18.42% ± 21.04% (grey; *n* = 3), which showed marginally greater but non-significant recovery for AR-13324 (*p* > 0.05). The last panel depicts the wound healing rates at the end of Week 1 (0–7 Days), Week 2 (7–14 Days), and Week 3 (14–21 Days), as well as the averaged wound healing rates for both scraped (black) and peeled (grey) wounds for (**E**) AR13324 and (**F**) Y27632.

**Figure 4 cells-12-01307-f004:**
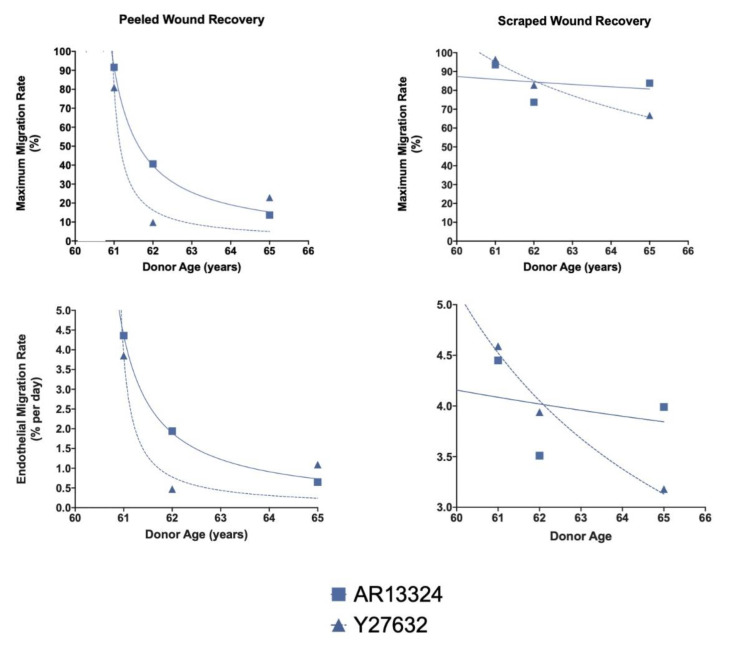
Effects of donor age on endothelial migration. Increasing age was associated with decreased maximum and daily endothelial migration rates for both peeled and scraped wounds. The correlation was stronger for peeled than scraped wounds for corneas treated with either Y-27632 (R^2^ = 0.98 and 0.87) or AR-13324 (R^2^ = 0.92 and 0.06), but without statistical significance.

**Figure 5 cells-12-01307-f005:**
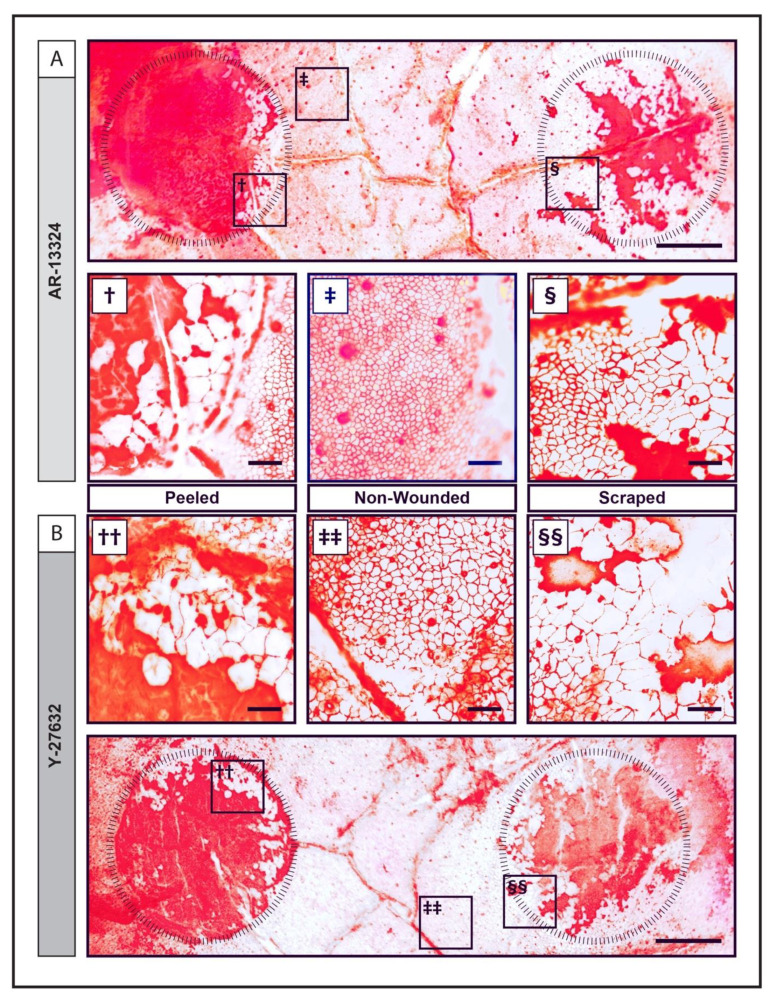
Histochemistry of corneas after endothelial wound regeneration. Alizarin red S staining of the same pair-matched corneas after 21 days in supplemented culture media with (**A**) AR-13324 and (**B**) Y-27632. The areas stained with vivid red represent the bare stroma and absence of CEC from the uptake of Alizarin Red dye. Scale bars: 1 mm. Ex vivo cornea wound recovery model of cornea treated with (**A**) AR-13324 and (**B**) Y-27632. Areas within the peeled areas can be seen with enlarged polygonal cells with clear cell membranes across the peeled wound margin for both AR-13324 (**^†^**) and Y-27632 (**^††^**). Insert of undamaged central endothelium. In both samples, the corneal endothelium showed “normal endothelial” features, as monolayered hexagonal cells, small cell body, and marked cell boundaries (**^‡^**,**^‡‡^**). There was no visible boundary of the scraped wound in either of the samples treated with AR-13324 (**^§^**) or Y-27632 (**^§§^**). Cells resembled corneal endothelium that showed a gradual increase in pleomorphism and as cells migrated towards the center of the wound. Scale bars: 100 μm.

**Table 1 cells-12-01307-t001:** Donor information for corneas procured for this study.

Serial Number	Age	Gender	Cell Count (OD/OS)	Cause of Death
01	26	Male	2398/2601	Testicular Cancer
02	15	Female	2809/2985	Multiple Blunt Force Injuries
03	4	Female	3623/2717	Anoxic Encephalopathy
04	18	Male	3257/3268	Subarachnoid Haemorrhage
05	31	Female	2985/3322	Acute Cardiac Crisis
06	14	Male	3021/3215	Drowning
07	28	Female	2833/2950	Suicide
08	35	Female	2513/2667	COPD/Cardiac Arrest
09	31	Female	2825/2653	Multi Vehicle Accident
10	20	Female	2538/2725	Multi Vehicle Accident
11	24	Female	2801/2849	Multi Vehicle Accident
12	30	Male	2950/3058	Multi Vehicle Accident
13	24	Male	3003/3236	Multi Vehicle Accident
14	33	Female	2825/2584	Gunshot Wound
15	24	Male	2976/3003	Multi Vehicle Accident
16	29	Male	3745/3953	Anoxic Brain Injury
17	27	Female	3146/3022	Postpartum Complications
18	19	Female	3364/3130	Complication of Liver Cancer
19	15	Female	3378/3106	Trauma
20	18	Male	3160/3253	Trauma
21	11	Female	2907/3040	Drowning
22	13	Male	3058/3175	Anoxia
23	66	Male	2421/2262	Brain Cancer
24	73	Female	2849/2681	Chronic Obstructive Pulmonary Disease
25	72	Male	2331/2513	Hypoxia
26	51	Male	2874/2398	Intracerebral Bleeding/Intracerebral Haemorrhage
27	63	Female	2387/2874	Acute Cardiac Event
28	69	Female	2053/2075	Chronic Obstructive Pulmonary Disease
29	64	Male	3077/3311	Sepsis
30	65	Male	2632/2778	Liver Cancer
31	61	Female	2577/2907	Breast Cancer
32	62	Male	2268/2326	Liver Failure

**Table 2 cells-12-01307-t002:** Summary of ROCK inhibitors used in this study.

ROCK Inhibitor Compounds	Compound Class	Concentration Range (Optimal)
Y-27632	Pyridine Carboxamide	10 µM
AR-13324 (Netasurdil)	Isoquinoline	100 nM to 1 µM
AR-13503	Isoquinoline	100 nM to 10 µM
AR-12286 (Verosudil)	Isoquinoline	100 nM to 10 µM
Y-39983	Pyrrolopyridine	100 nM
K-115 (Ripasudil)	Isoquinoline	30 µM
G-1	Indazole	30 µM
G-2	Aminofurazan	10 µM
G-3	Amide	30 µM
G-4	Benzodiazapiene	10 µM

## Data Availability

The data presented in this study are available on request from the corresponding author.

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
