# Peer review of "Effects of Rho-Associated Kinase (Rock) Inhibitors (Alternative to Y-27632) on Primary Human Corneal Endothelial Cells"

_cells, 2023, doi:10.3390/cells12091307_

Round 1

Reviewer 1 Report

Peh et al., in their manuscript titled “Effects of alternative rho-associated kinase (rock) inhibitors on  corneal endothelial cells” have investigated the role of the active ingredient of Netarsudil on endothelial cell attachment, proliferation and regeneration on exvivo cornea using wound closure model. The three ROCK inhibitors in clinical trials are Netarsudil (AR-13324), Verosudil (AR-12286) and Ripasudil (K-115) in glaucoma. Moreover, AR-13324 is a FDA approved drug.  The study findings have implication in the corneal endothelial transplantation studies. There are few concerns that need to be addressed:

1   1. The xCelligence experiment shows that Netarsudil at higher concentrations does not show any recovery in recovery phase, whereas the active compound AR-13503 at higher concentration in the recovery phase shows better attachment comparable to Y-27632. Can the authors provide a plausible explanation for the difference in the cell attachment studies? In similar lines, the authors also show a comparable rate of proliferation at lower concentration (1um and 100nm) of AR-13324 and 10um of Y-27632. Here the authors show higher rate of proliferation on increasing the concentration of AR-13503 (active compound). It would be helpful, if the authors explain the reason for various concentrations used in the different experiments? For attachment assays, 100nm, 10um, 30 um; for proliferation assay, 100nm, 1um, 10um and then for wound closure assay 1um and 10um??

    2.  It is unclear as to how the authors conducted the experiment wherein they created the wound in day 0 and also observed recovered areas on day 0 which roughly around 50%. Please explain the experimental design? Why is there no untreated cornea to define a baseline recovery? Also in the attachment experiments (xCelligence ) it would be interesting to note the attachment efficiency of endothelial cells without the ROCKi.

33.The active component of Netrasudil (AR-13503) shows better attachment and proliferation, but the authors have not shown the cornea wound recovery study with AR-13503, but instead investigate the role of AR-13324. Based on what basis the authors selected AR-13324 or AR-13503 to be investigated?

44.It always good to have multiple experiments to validate or conclude a finding. The authors have used a single technique to cell attachment, proliferation and wound closure. It would be nice to see the physiological readouts supported by molecular or protein studies. This might explain several mitigating components dictating the outcome such as wound closure, where the authors don’t see any end point difference though the speed of recovery differs between Y-27632 and AR-13324. Please show some staining such as proliferation markers, EMT markers, apoptosis markers to support the findings.

55. The authors are interested in the clinical applicability and regenerative translational study. The concentration window for AR-13324 is very small whereas the same for Y-25632 is wider. It would be interesting for the readers to know the authors take based on the differential concentration window of the ROCKi and its relevance with respect to the translational regenerative medicine in corneal endothelial cell studies.

66. Please change “FCED” to “FECD” throughout the manuscript. It is Fuchs Endothelial Corneal Dystrophy and not Fuchs Corneal Endothelial Dystrophy.

Reviewer 2 Report

This is an extensive study on the efficacy of rho inhibitors vis-a-vis endothelial health and wound healing. The writing needs to be clarified at various points. For example, some measurements could be better explained with positive/negative controls. In addition, the science behind the pharmacological effects of the Rho kinase inhibitors must be sufficiently well described.

Line 2 - Title – alternative to rho-kinase inhibitors – the word alternative does not make sense without mentioning Y-27632. Please complete the title more effectively.

Lines 21-22 - I would be surprised if others have not examined this in the literature.

Line 23 – how do you distinguish between “optimal” and “effective”? They appear identical to me.

Line 31 – a typographical error

The abstract is lengthy and can be shortened. However, the abstract also does mention any data on the statistical significance of the results.

Line 51- what is meant by “rich in tight junctions”? The corneal endothelium is a leaky epithelium since the number of tight junctional strands are few.

Line 55 – should be “endothelial.” Also, please avoid the abbreviation CED as it is not standard.

Line 101-105 – Provide references.        

Line 47-84: It would be good to abridge these lines. The Introduction is very long.

Line 167 – Are we sure that In silico experiments described here have not been performed before? Explain if your computations/simulations are new or different. I am also surprised that there are citations in the paragraph.

Line 198 – Describe xCelligence in 3-4 lines before detailing the experiments. Otherwise, the reader will not understand what was measured here. You talk about impedance in Line 216. Also, provide positive and negative control for cell index. What drugs/agents increase/decrease cell index?

Line 221: The details of the xCELLigence real-time cell analyzer (ACEA Biosciences) are repeated.

Line 226 –manufacturer’s name is repeated in the line. Not necessary.

Lines 234-236 – Description of Click-iT Cell Proliferation assay should be Line 231 before discussing the two ROCK inhibitors.

Line 233 – Cell index does NOT reach 1.0 when AR-13503 was 1 micromolar (fig. 1). What is the relevance?

Could you remove the subheadings in lines 270 and 278?

Correct units for concentration are in Fig. 2. Statistics are not indicated.

Line 333: Table 2 shows less than 10 drugs but not 46 compounds.

Line 342-343. I cannot understand the sentence. A typical reader would need help understanding the docking score. Could you define/explain the term before using it? How does it relate to affinity?

Fig. 3 – In the legend, indicate the number of biological repeats. Also, show statistical comparisons.

Line 352. What determines the impedance? Explain before using the term. Also, note that the Y axis in Fig. 2 is in terms of cell index. Importantly, you have yet to show control agents that can increase or decrease cell index.

Fig. 4 – lacks a significant number of biological repeats.

Lines 462-464 – Please double-check for accuracy. Does not seem right to me.

Lines 455-468 – Are you correlated to cell adhesion to cell index/impedance?

Round 2

Reviewer 2 Report

authors have improved the m/s and responded to all criticisms point-by-point. t